# Shadow Enhancement Using 2D Dynamic Stochastic Resonance for Hyperspectral Image Classification

**Qiuyue Liu** [1,2], **Min Fu** [3] **and Xuefeng Liu** [1,2,*]

1    College of Automation and Electronic Engineering, Qingdao University of Science and Technology, Qingdao 266061, China
2    College of Electromechanical Engineering, Qingdao University of Science and Technology, Qingdao 266061, China
3    College of Information Science and Engineering, Ocean University of China, Qingdao 266100, China
*    Correspondence: snowclub@qust.edu.cn

**Abstract:** With the development of remote sensing technology, classification has become a meaningful way to explore the rich information in hyperspectral images (HSIs). However, various environmental factors may cause noise and shadow areas in HSIs, resulting in weak signals and difficulties in fully utilizing information. In addition, classification methods based on deep learning have made considerable progress, but features extracted from most networks have much redundancy. Therefore, a method based on two-dimensional dynamic stochastic resonance (2D DSR) shadow enhancement and convolutional neural network (CNN) classification combined with an attention mechanism (AM) for HSIs is proposed in this paper. Firstly, to protect the spatial correlation of HSIs, an iterative equation of 2D DSR based on the pixel neighborhood relationship was derived, which made it possible to perform matrix SR in the spatial dimension of the image, instead of one-dimensional vector resonance. Secondly, by using the noise in the shadow area to generate resonance, 2D DSR can help increase the signals in the shadow regions by preserving the spatial characteristics, and enhanced HSIs can be obtained. Then, a 3DCNN embedded with two efficient channel attention (ECA) modules and one convolutional block attention module (CBAM) was designed to make the most of critical features that significantly affect the classification accuracy by giving different weights. Finally, the performance of the proposed method was evaluated on a real-world HSI, and comparative studies were carried out. The experimental results showed that the proposed approach has promising prospects in HSIs' shadow enhancement and information mining.

**Keywords:** remote sensing image; DSR; shadow area; CNN; attention mechanism

## 1. Introduction

Owing to remote sensing technology's rapid growth, hyperspectral images (HSIs) containing rich information in the spatial and spectral dimensions [1] have extensive use in agricultural production [2], urban planning [3], environmental monitoring [4], and so on. By assigning category labels to each image pixel according to the sample characteristics, classification has become one of the effective means to extract information from HSIs [5].

Classical classification methods, including support vector machine (SVM) [6], k-nearest neighbor (K-NN) [7], maximum likelihood estimation (MLE) [8], dimension-reduction-based methods [9], linear discriminant analysis (LDA) [10], independent component analysis (ICA) [11], principal component analysis (PCA) [12], etc., have been used for HSI classification with good properties. However, they either need to reduce the dimensions of the data or can only obtain shallow features. In recent years, deep learning with a strong performance in extracting nonlinear features has been successfully applied to hyperspectral data processing [13]. The principle of deep learning classification is to extract features from basic to deep without pre-designing features. As an effective feature extraction method in image processing, convolutions can help obtain a feature map of an image after the

convolutional operations. By arranging and combining low-level features at a higher level from the input layer to the output layer of the network, the features of an image are continuously extracted and abstracted, and classification and recognition can be achieved based on these features ultimately [14]. Typical networks such as the deep belief network (DBN) [15] and stacked auto-encoder (SAE) [16] can obtain deep features by layered training on the premise that the input should be converted to a one-dimensional vector. Besides, there are studies and achievements based on the graph neural network (GNN) designed for and targeted at irregular data, i.e., social networks and molecular networks [17,18].

Frameworks evolved from convolutional neural networks (CNNs), such as generative adversarial networks (GANs) [19] and Res-Net [20], can extract features with the consideration of both the spatial and the spectral information [21]. However, many feature maps within a layer show much pattern similarity, so that these features could be redundant, which means that, if one model has extracted information from one feature map, it will only need to extract the difference from its similar ones [22]. Meanwhile, on account of the single structure of these frameworks, all features are considered of equal importance, and some critical features influencing the classification effect significantly have not been made full use of. Therefore, the attention mechanism has emerged and is introduced to the convolutional neural network to evaluate the importance of the extracted features to ensure that the essential features are taken seriously enough in the classification [23,24].

However, due to the influence of cloud cover, light, and other environmental factors, some HSIs contain shadow areas, in which signals are weakened and information extraction, including classification, is difficult [25]. The HSIs spatial–spectral information enhancement can bring positive effects to HSIs' classification [26,27]. The conventional approaches in the spatial domain and the transform domain for image enhancement [28,29], such as Retinex [30], histogram correction [31], the low pass filter (LPF) [32], the autoregressive moving average (ARMA) filter [33], and so on, mainly focus on removing noise, which would inevitably result in the loss of some signals and may destroy the correlation of the data [34]. There are methods based on neural networks for image enhancement [35] with time-consuming calculations and insufficient samples. Currently, the exploration of rich information in the shadow of hyperspectral images is still a difficult point in existing research. Using resonance generated by noise to improve the signal, one-dimensional dynamic stochastic resonance (DSR) has been introduced to enhance the signal with noise [25]. However, a two-dimensional spatial image must be converted to a one-dimensional vector before being processed by 1D DSR, which will inevitably destroy the spatial correlation of the image.

Therefore, an iterative equation of 2D DSR was derived in this paper, which can protect the spatial correlation of HSIs by performing matrix SR in the spatial dimension of the image instead of one-dimensional vector resonance. Furthermore, to fully utilize critical features that affect the classification result substantially, a 3DCNN embedded with two efficient channel attention (ECA) modules and one convolutional block attention module (CBAM) is proposed. By distributing the corresponding weights to the feature maps of the input, the attention mechanism can help the model make estimates more accurate without more consumption of the storage and computation of the model. The performance of the proposed shadow enhancement and classification approach was verified on a real-world HSI.

In this paper, on the one hand, the derivation of 2D DSR can not only develop the signal enhancement ability, but also protect the spatial correlation of image signals, laying the foundation for further information extraction. On the other hand, with two efficient channel attention modules and one convolutional block attention module, an improved 3DCNN was designed to fully utilize the key features and increase the classification performance.

The remaining content of this paper is organized as follows: Section 2 introduces the basic theories of the proposed technique, including the principles of the nonlinear bistable DSR system, the basic construction of the CNN, the derivation of 2D DSR based on the pixel neighborhood relationship, and the structure of the CNN with attention modules;

Section 3 introduces the experimental details and results; Section 4 presents the comparison and discussion; Section 5 gives the conclusion.

## 2. Materials and Methods

### 2.1. Dynamic Stochastic Resonance

In image processing, noise affects the quality of the image and usually needs to be removed to increase the signal-to-noise ratio (SNR) of the image. However, the valuable signals in the image, especially the correlation between the signals, might inevitably be destroyed by the noise reduction when the noise spectrum is close to the signal spectrum. In some specific nonlinear systems, with stochastic resonance occurring, internal or external noise can help to enhance weak signals because some noise energy can be transferred into signal energy, and then, the SNR of the system output can be increased [36,37]. DSR is a spatial domain analysis method that correlates the bistable system parameters of the double-well potential with the intensity value of the noisy image.

On the basis of Langevin's equation of motion, the 1D nonlinear expression of the overdamped dynamic system can be [38]

$$\frac{dh(t)}{dt} = -\frac{dH(h)}{dh} + I(t) + \mu(t) \tag{1}$$

where $I(t)$ is the periodic input signal, $\mu(t)$ is the intensity distribution of the noise, and $t$ and $h(t)$ are the time and spatial location of a particle moving in a bistable potential well. $H(h)$ is the potential function affected by the displacement:

$$H(h) = -\frac{1}{2}ah^2 + \frac{1}{4}bh^4 \tag{2}$$

where $a$ and $b$ are system parameters. Figure 1 plots the situation $a = b = 2$.

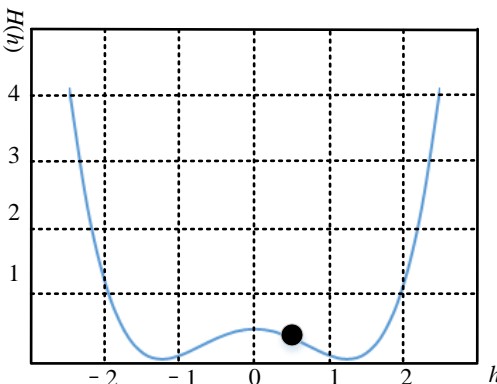

**Figure 1.** Situation of the system with $a = b = 2$.

Substitute Equation (2) for $H(h)$ in Equation (1), then

$$\frac{dh(t)}{dt} = ah(t) - bh^3(t) + I(t) + \mu(t) \tag{3}$$

where $x\pm = \pm\sqrt{\frac{a}{b}}$ are the two stable points in Equation (3) and $\Delta H = \frac{a^2}{4b}$ is a barrier of the system. If the periodic driving force, i.e., the periodic input signal $I(t)$, is absent, the system remains stable. If a periodic force is imposed on the bistable system, the system's stability will be damaged and periodic changes will occur in the potential well. By cooperating with the periodic driving force, noise can provide energy for the particles to transform in two stable states. In other words, noise can help the signal obtain higher energy in the stochastic resonance system.

### 2.2. Convolutional Neural Network

With a deep structure including convolutional calculation, the CNN has been extensively applied to text, voice, image, video processing, and pattern recognition [39]. Based on the matrix-weight-sharing structure, representation learning ability, and shift invariance, the CNN has become a suitable model for processing HSI data [40]. The main structure of the CNN, including the input, convolutional, pooling, fully connected (FC), and output layers, is illustrated in Figure 2.

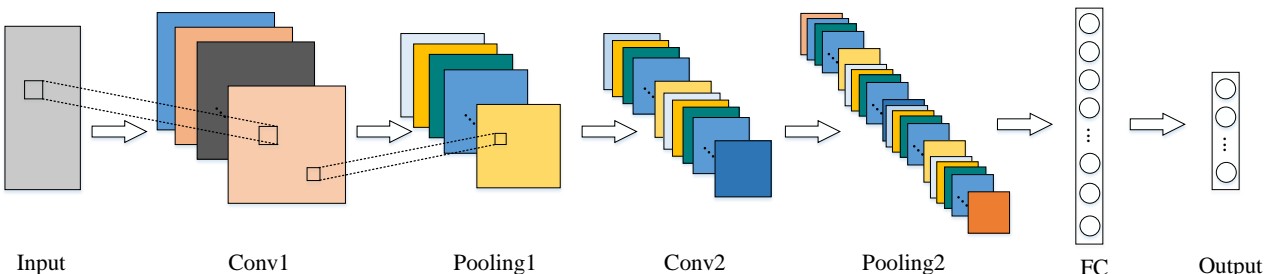

**Figure 2.** The main structure of the CNN. Conv1 and Conv2 represent Convolutional Layers 1 and 2. Pooling1 and Pooling2 stand for Pooling Layers 1 and 2.

The input layer of the CNN can process multidimensional data. In the convolutional layers, convolutional kernels can extract features from the input data, and the activation function can make it easier to express complicated features. The convolution operation can be expressed as [41]

$$CH_n^l = A(\sum_{m=1}^{M} \sum_{n=1}^{N} CH_m^{l-1} s_n^l + r_n^l) \tag{4}$$

where $CH_n^l$ is the $n$-th characteristic matrix of the $l$-th layer, $A(\cdot)$ indicates the activation function, M and N represent the number of neurons in the last and the current layer, respectively, and $s_n^l$ and $r_n^l$ are the weight matrix and offset of the corresponding convolution kernel.

After feature extraction, the output feature maps are transferred to the pooling layer to select features and filter information. The pooling layer can decrease the dimension of the feature map by downsampling, which can significantly cut down the number of neurons and the computational difficulty of the network.

By using the existing higher-order features, the fully connected layer can be combined with the nonlinear extracted features to gain the output. The feature map can be expanded into vectors in a fully connected layer by connecting each neuron with all the neurons in the previous layer.

The logic or softmax activation function is generally used to output classification labels for image classification. The commonly used softmax function can be expressed as

$$g(Out_{FC}) = softmax(wgOut_{FC} + Offset) \tag{5}$$

where $w$ and $Offset$ are the vectors of the weight and offset and $Out_{FC}$ is the output of the FC layer.

Currently, the CNN has different convolution kernels according to the dimension of the input data, including 1-dimensional (1D), 2D, and 3D, which have the same element calculation process and adopt backpropagation to modify the parameters.

### 2.3. Two-Dimensional DSR Shadow Enhancement for Hyperspectral Image Classification by CNN Embedded with Multiple Attention Mechanisms

Because of light, cloud cover, and other environmental factors, there are shadow regions in some HSIs where signals are weak and information can hardly be analyzed.

Meanwhile, all features extracted by the classification methods based on deep learning are generally considered equally important, so a few key features that seriously affect the classification result cannot be effectively made use of. Therefore, an iterative equation of 2D DSR was derived to enhance the signal in shadow areas, and a 3DCNN embedded with multiple attention mechanisms (MAM-3DCNN) is proposed for HSI classification in this paper. The proposed approach's main procedure is shown in Figure 3.

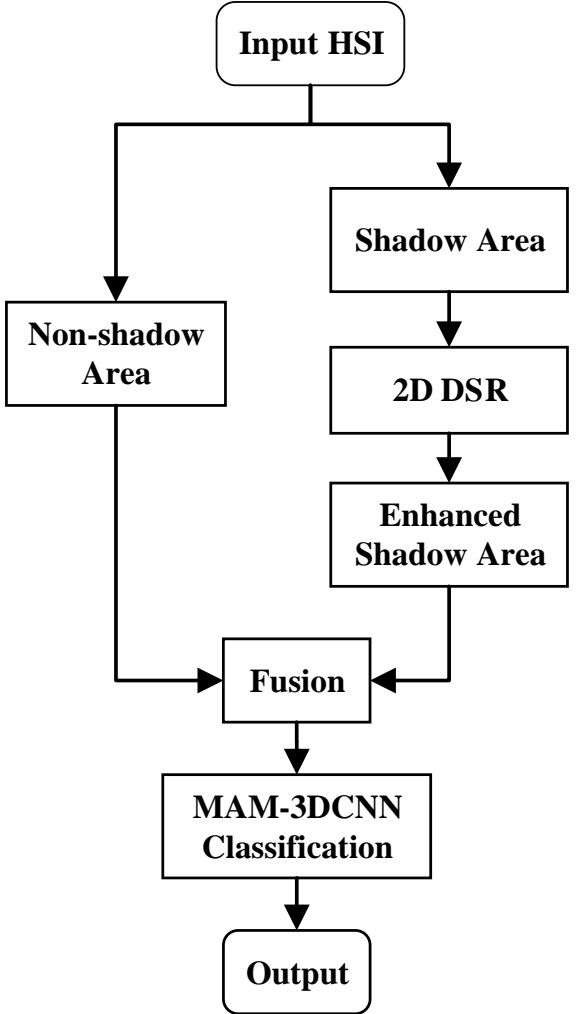

**Figure 3.** The main procedure of the proposed approach.

2.3.1. Two-Dimensional Dynamic Stochastic Resonance

Classical stochastic resonance theory mainly deals with 1-dimensional vectors, so 2-dimensional and multidimensional data must be converted to 1 dimension before resonance processing. To preserve the spatial correlation of HSIs, an iterative equation of 2-dimensional DSR carrying out stochastic matrix resonance in the spatial dimension was derived.

For the shadow areas in HSIs, there are both a weak signal $\hat{I}(x,y)$ and noise $\hat{\mu}(x,y)$ with $x$ and $y$ being the spatial position of the pixel. Denoting $\hat{f} = \hat{I}(x,y) + \hat{\mu}(x,y)$ and according to the 1D DSR in Equation (1), a 2D nonlinear expression of the bistable stochastic resonance system can be

$$\frac{\partial h^2(x,y)}{\partial x \partial y} = -\theta\left(\frac{\partial h}{\partial x} + \frac{\partial h}{\partial y}\right) + \hat{a}h(x,y) - \hat{b}h^3(x,y) + \hat{f}(x,y) \qquad (6)$$

where $h(x,y)$ is the system output and $\theta$ ($> 0$) is the damping term of the system.

If the damping term $\frac{\partial h^2(x,y)}{\partial x \partial y}$ is large, the second-order term in Equation (6) can be ignored [42], and Equation (6) can be rewritten as

$$0 = -\left(\frac{\partial h}{\partial x} + \frac{\partial h}{\partial y}\right) + \frac{\hat{a}}{\theta}h(x,y) - \frac{\hat{b}}{\theta}h^3(x,y) + \frac{\hat{f}(x,y)}{\theta} \tag{7}$$

By replacing $\frac{\hat{a}}{\theta}$, $\frac{\hat{b}}{\theta}$, and $\frac{\hat{f}(x,y)}{\theta}$ with $a$, $b$, and $f(x,y)$, respectively, Equation (6) can be simplified as the following overdamped partial differential equation:

$$\frac{\partial h(x,y)}{\partial x} + \frac{\partial h(x,y)}{\partial y} = ah(x,y) - bh^3(x,y) + f(x,y) \tag{8}$$

Based on the characteristic line theory [43], Equation (8) can be equivalent to two ordinary differential equations:

$$\begin{cases} \frac{dx}{1} = \frac{dh}{ah - bh^3 + f} \\ \frac{dy}{1} = \frac{dh}{ah - bh^3 + f} \end{cases} \tag{9}$$

where the characteristic line here is $\frac{dx}{1} = 1$, i.e., $y = x + C$ with $C$ being a constant. Therefore, in any small neighborhood, the interpretation of Equation (9) is symmetric about the diagonal direction [43], and solving the equations can be equivalent to independently solving the following equations:

$$\begin{cases} \frac{dh}{dx} = ah - bh^3 + f(x) \\ \frac{dh}{dy} = ah - bh^3 + f(y) \end{cases} \tag{10}$$

The equivalent difference form of Equation (10) can be expressed as [44]

$$\begin{cases} h_{i,j,k} = t_x[ah_{i,j-1,k} - bh_{i,j-1,k}^3 + f_{i,j-1,k}] + h_{i,j-1,k} \\ h_{i,j,k} = t_y[ah_{i-1,j,k} - bh_{i-1,j,k}^3 + f_{i-1,j,k}] + h_{i-1,j,k} \end{cases} \tag{11}$$

where $i$, $j$ are the abscissa and ordinate positions in the spatial dimension of the input shadow data $f$, $k$ represents the $k$-th band of the HSI, i.e., $f_{i,j-1,k}$ is the pixel at the $(i, j-1)$ spatial position on the $k$-th band, $t_x$ and $t_y$ represent the sampling interval in the direction of abscissa and ordinate, respectively, and $h_{i,j,k}$ is the output of the system at $(i, j)$ on the $k$-th band.

Since Equation (11) means nonlinear filtering of the input into horizontal and vertical directions at the same time, it can be extended to a four-way parallel difference form with the iterative update:

$$\begin{cases} h_{i,j,k}(n+1) = t_x[ah_{i,j-1,k}(n) - bh_{i,j-1,k}^3(n) + f_{i,j-1,k}] + h_{i,j-1,k}(n) \\ h_{i,j-1,k}(n+1) = t_x[ah_{i,j,k}(n) - bh_{i,j,k}^3(n) + f_{i,j,k}] + h_{i,j,k}(n) \\ h_{i-1,j,k}(n+1) = t_y[ah_{i,j,k}(n) - bh_{i,j,k}^3(n) + f_{i,j,k}] + h_{i,j-1,k}(n) \\ h_{i,j,k}(n+1) = t_y[ah_{i-1,j,k}(n) - bh_{i-1,j,k}^3(n) + f_{i-1,j,k}] + h_{i-1,j,k}(n) \end{cases} \tag{12}$$

where $n$ indicates the number of iterations. The iterative process of Equation (12) on the $k$-th band of the HSI by 2D DSR is shown in Figure 4. Each pixel's output combines spatial information in the upper, lower, left, and right directions, so that the correlation between the spatial pixels can be maintained.

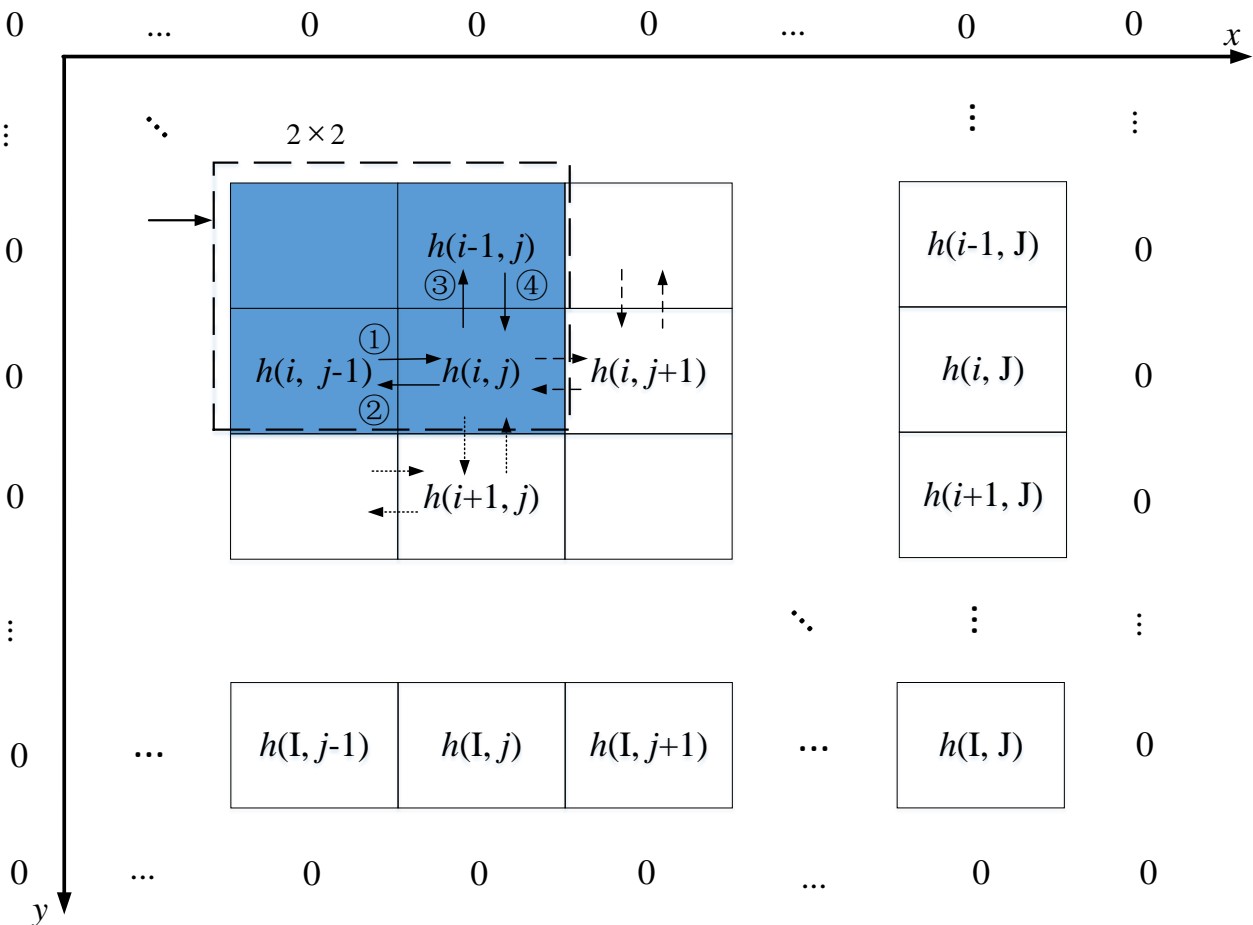

**Figure 4.** The iterative process of 2D DSR on the *k*-th band of the HSI. A 2 × 2 window is used to slide the sampling. The dashed lines of different types represent relevant pixels involved in updating the enhancement value of a pixel through Equation (11). The ①,…, ④ correspond to the relationship between the pixels in the first to fourth sub-formulas in Equation (12). *I* × *J* represents the size of the data in each band of the HSI.

### 2.3.2. Three-Dimensional Convolutional Neural Network with Multiple Attention Mechanisms

To exert the potential value of critical features that significantly impact the classification results, multiple attention modules were embedded into a 3DCNN. On the one hand, as a local cross-channel interaction strategy without dimension reduction, ECA can significantly improve network performance and avoid the adverse effect of compression and dimensionality reduction on the dependence between learning channels [45]. On the other hand, as a lightweight general module, a CBAM can be integrated into any CNN by adding a channel attention (CA) mechanism and a spatial attention (SA) mechanism to emphasize the channel and spatial characteristics [46]. In addition, HSIs have spectral data in hundreds of dimensions, resulting in complex channel states in the network, so double ECAs were inserted in the CNN and proven to be more effective in learning channel attention than one ECA through experiments. The main structure of the proposed MAM-3DCNN for HSI classification is illustrated in Figure 5.

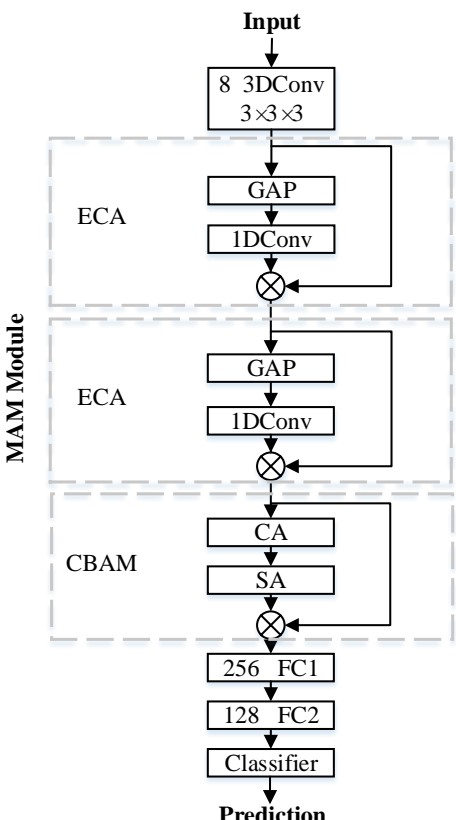

**Figure 5.** The main structure of the proposed MAM-3DCNN. The size of the convolutional kernel is $3 \times 3 \times 3$, and the 8 3DConv means 3D convolution with 8 convolutional kernels. GAP is the global average pooling. $\otimes$ denotes the positionwise dot product. The 256 FC1 is the 1st fully connected layer with 256 neurons.

Based on the aggregated features obtained by the global average pooling, ECA can generate channel weights by a fast 1D convolution, and the specific structure is illustrated in Figure 6.

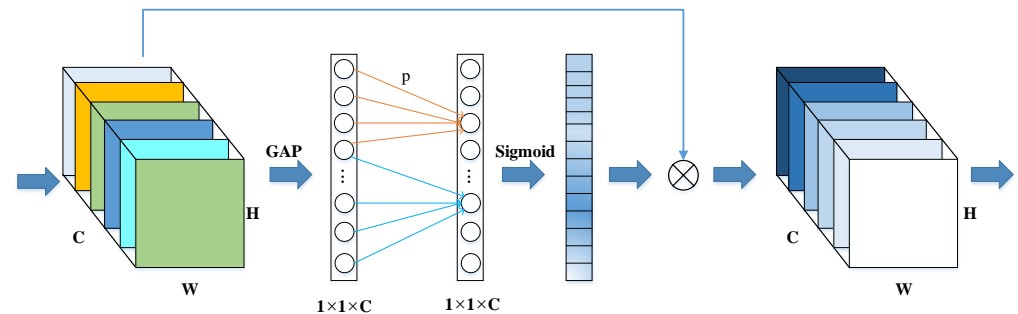

**Figure 6.** The specific structure of ECA. The input feature maps' height and width are represented by H and W, and C is the number of channels. *p* represents the required adjacent channels to obtain the cross-channel interaction information of each channel and can be adaptively determined via a mapping of C. Sigmoid is the activation function.

The convolutional block attention module is composed of spatial attention and channel attention in series, as shown in Figure 7. In the CA of the CBAM shown in Figure 8, the global average pooling (GAP) and global maximum pooling (GMP) are firstly used to aggregate the spatial information of the input feature map, then the number of channels is

compressed to C/r, with r being the compression ratio to reduce the parameter overhead, and the channel feature vectors can be obtained by elementwise summation finally. As a supplement to the channel attention, in the SA, the average pooling and maximum pooling are performed and an effective feature descriptor is generated by connecting the results, then a standard convolutional layer connection and convolution are combined to generate a 2D spatial attention map.

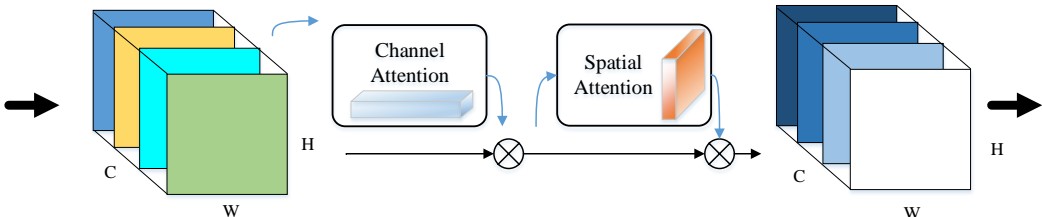

**Figure 7.** The basic structure of the CBAM. The channel attention module assesses the importance of each channel and gives the input channels the corresponding weights, and the spatial attention offers different attention to pixels in each channel according to the significance.

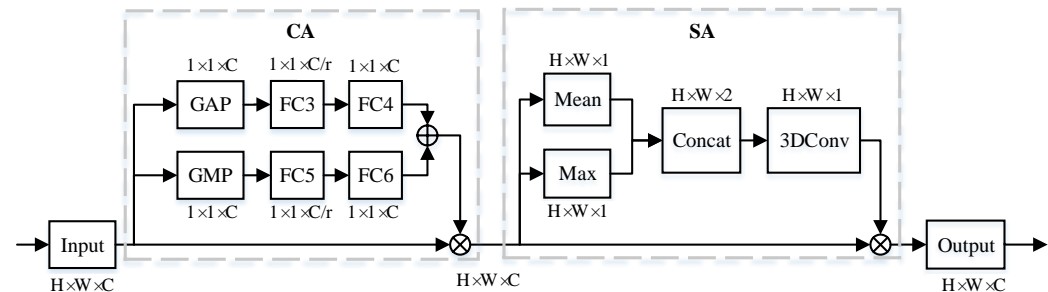

**Figure 8.** The detailed structure of the CA and SA in the CBAM. The $H \times W \times C$ represents the data with the H height, W width, and C channels. $r$ is the compression ratio. $\oplus$ denotes elementwise summation. Mean and Max represent the average pooling and maximum pooling. Concat means feature fusion.

### 2.3.3. The Procedure of the Proposed MAM-3DCNN

In actual HSIs' processing, the HSIs with shadows are three-dimensional data with a length (L), width (W), and height (H), and this can be represented as $M^{H \times W \times L}$, where L represents the number of bands in the spectral dimension. The 2D data in each band can be defined as $\chi = [X^1, X^2, \dots, X^L]$, and the pixel at (p,q) in the b-th band is $X_{p,q}^b$. Therefore, the specific steps can be carried out in detail:

Step 1: Firstly, a shadow mask needs to be set to extract the shadow area $\chi_{sd} = [X_{sd}^1, X_{sd}^2, \dots, X_{sd}^L]$ in the HSI.

Step 2: For the extracted data, 2D DSR in Equation (12) can be applied to each band of the data $X_{sd}^b$ to enhance each pixel $X_{sd(p,q)}^b$ by making use of the spatial information in the neighborhood, and then, the enhanced shadow data $\chi_{sd-ed}$ can be obtained.

Step 3: By fusing the enhanced shadow data with non-shadow data, the 2D DSR-enhanced HSI $\chi_{enhanced}$ can be acquired.

Step 4: To reduce the impact of unrelated information and computing costs, the most-important 10 components are extracted by principal component analysis (PCA) before classification.

Step 5: Finally, the dimensionality reduced data are divided according to the window size as the constructed classification network's input to obtain the final result.

## 3. Experiment

To assess the performance of the proposed approach, a real-world HSI dataset was used and the experiments were carried out in the Ubuntu 16.04 operating system on an NIVIDIA GTX2080Ti with 11GB memory. The programs and models were built on Keras.

### 3.1. Dataset

The real-world Hyperspectral Digital Imagery Collection Experiment (HYDICE) data with a shadow area were adopted in this paper. With a 0.75 m spatial resolution and a 10nm spectral resolution, it consists of 316 rows, 216 columns, and 148 spectral bands from 435 to 2326 nm. Figure 9 lists the original HYDICE image and the ground truth. The labels and the number of samples are presented in Table 1. In the classification experiments, the HYDICE data were randomly divided into training and test subsets without overlap, with the test subset accounting for 80%.

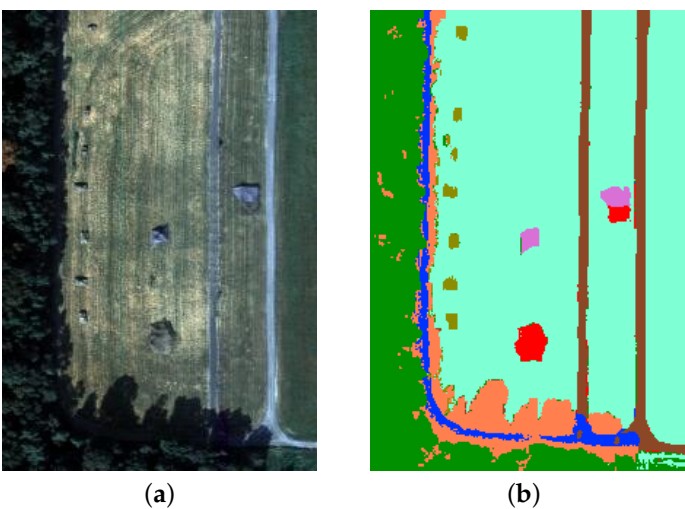

| (a) | (b) |

**Figure 9.** The original HYDICE and ground truth. (**a**) HYDICE. (**b**) Ground truth.

**Table 1.** Information of the ground truth. The labels, sample number, and represented colors for each category in the ground truth are displayed.

| Number | Color | Sample | Label |
|--------|-------|--------|-------|
| 1 | | 33,184 | Grass |
| 2 | | 10,850 | Tree |
| 3 | | 3376 | Road |
| 4 | | 1686 | Road in shadow |
| 5 | | 323 | Grass in shadow |
| 6 | | 537 | Target 1 |
| 7 | | 514 | Target 2 |
| 8 | | 4135 | Target 3 |

To compare the classification results quantitatively, as commonly used evaluation indices for HSIs, the overall accuracy (OA), average accuracy (AA), and Kappa coefficients were introduced to the experiments [47]. The OA can be defined as

$$OA = \frac{\sum\limits_{u}^{U} R_{u,v}}{G} \times 100\% \tag{13}$$

where $U$ indicates the number of labels, $R$ with the size of $U \times U$ is the confusion matrix, $R_{u,v}$ represents samples belonging to label $u$, but misclassified into label $v$, and $G$ is the number of the tested samples.

Correspondingly, the *AA* and *Kappa* coefficients can be expressed as below:

$$AA = \frac{1}{U} \sum_u^U \frac{R_{u,u}}{\sum_v^U R_{u,v}} \times 100\% \qquad (14)$$

$$Kappa = \frac{G \sum_u^U R_{u,v} - \sum_u^U \sum_v^U (R_{u,v} \times R_{v,u})}{G^2 - \sum_u^U \sum_v^U (R_{u,v} \times R_{v,u})} \times 100\% \qquad (15)$$

Besides, the classification accuracy of each category was calculated by the recall value, which indicates the correctly predicted positive samples and is calculated as:

$$Recall = \frac{R_{u,u}}{\sum_v^U R_{u,v}} \qquad (16)$$

*3.2. Parameter Setting*

3.2.1. Setup of 2D DSR Parameters

The setting of the parameters in DSR directly affects the depth of the potential well, the barrier and the vibration state of the system, etc. However, there is no straightforward algorithm to determine the parameter values of a specific given resonance system, and in practice, the values are determined numerically by fitting various applications. The number of parameters to be set in 2D DSR increases from 2 in 1D DSR to 5, which are $a$, $b$, $t_x$, $t_y$, and $n$ in Equation (12).

Since the purpose of DSR enhancement in this paper was to improve the classification accuracy of ground targets in shadow areas, the OA was used to measure the rationality of the parameter settings. With $a$, $b$, $t_x$, and $t_y$ from 0 to 5 and $n$ from 1 to 10, the enhanced image can be acquired by fusing the shadow area processed by 2D DSR with the non-shadow area and then classified by the 3DCNN. According to the above principle, the appropriate parameter for the HYDICE image in Figure 9 can be set as $t_x = t_y = a = b = 0.01$ and $n = 5$. The OA values under different $n$ are shown in Figure 10.

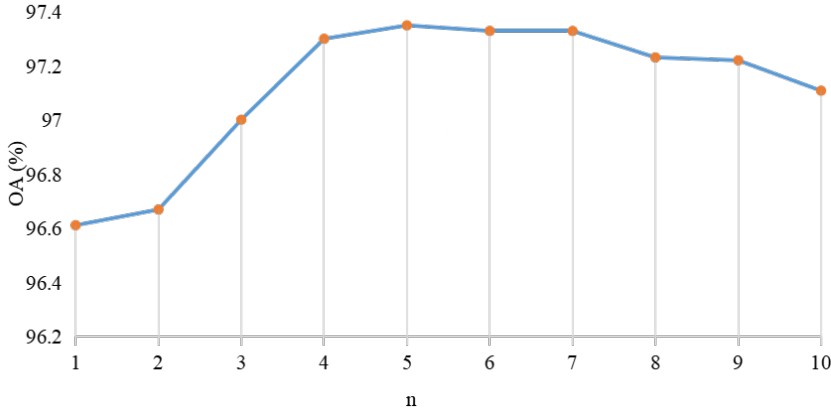

**Figure 10.** OA values under different $n$. The best result is obtained when $n = 5$.

3.2.2. Parameter Setting of MAM-3DCNN

In the experiments, to obtain a stable network performance, the internal parameters of the MAM-3DCNN in Figures 5–8 were set as shown in Table 2, and the configuration of the network operation is displayed in Table 3. Moreover, through network debugging, setting the parameters $p$ in Figure 6 to 3 and $r$ in Figure 8 to 1 was suitable for the HYDICE

HSI. The OA, AA, and Kappa values under different parameter settings of *r* are shown in Figure 11.

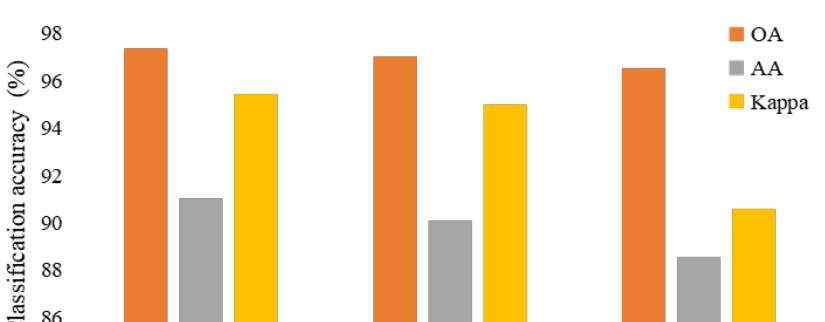

**Figure 11.** Comparison of classification accuracy under different *r*s. The best result of the OA, AA, and Kappa can be obtained when *r* = 1.

**Table 2.** Internal parameter setting of the MAM-3DCNN.

| | Layer | | Kernel | Kernel Size | Activation | Dropout |
|---|---|---|---|---|---|---|
| | 3DConv | | 8 | $3 \times 3 \times 3$ | Relu | - |
| MAM | ECA1 | 1DConv | 1 | $p = 3$ | Sigmoid | - |
| | ECA2 | 1DConv | 1 | $p = 3$ | Sigmoid | - |
| | CBAM | FC3/FC5 (3DConv) | 8 | $1 \times 1 \times 1$ | Relu | - |
| | | FC4/FC6 (3DConv) | 8 | $1 \times 1 \times 1$ | - | - |
| | | 3DConv | 1 | $3 \times 3 \times 3$ | Sigmoid | - |
| | FC1 | | 256 | - | Relu | 0.6 |
| | FC2 | | 128 | - | Relu | 0.5 |

**Table 3.** Configuration of the network operation.

| Name | Setting |
|---|---|
| Window size | 11 |
| Test ratio | 0.8 |
| Learning rate | 0.001 |
| Optimizer | Adam |
| Epoch | 100 |
| Loss function | Categorical cross-entropy |

### 3.3. Experimental Results

3.3.1. Shadow Enhancement by 2D DSR

Theoretically, the expansion of DSR from 1D to 2D can maintain the spatial correlation of the HSI data, so the effect of shadow enhancement by 1D DSR [25] and 2D DSR was focused on in this paper. Firstly, the HYDICE data were normalized to meet the small parameter requirements of DSR. Secondly, a shadow mask constructed from the ground truth in Figure 8b was applied to acquire the shadow data in HYDICE. Then, each band of the extracted image can be processed by 2D DSR in Equation (12). Finally, the HSI with an enhanced shadow area could be attained by fusing the enhanced shadow area with the original image. The results of HYDICE enhanced by 1D and 2D DSR are shown in Figure 12, and the classification accuracy of 3DCNN is illustrated in Table 4.

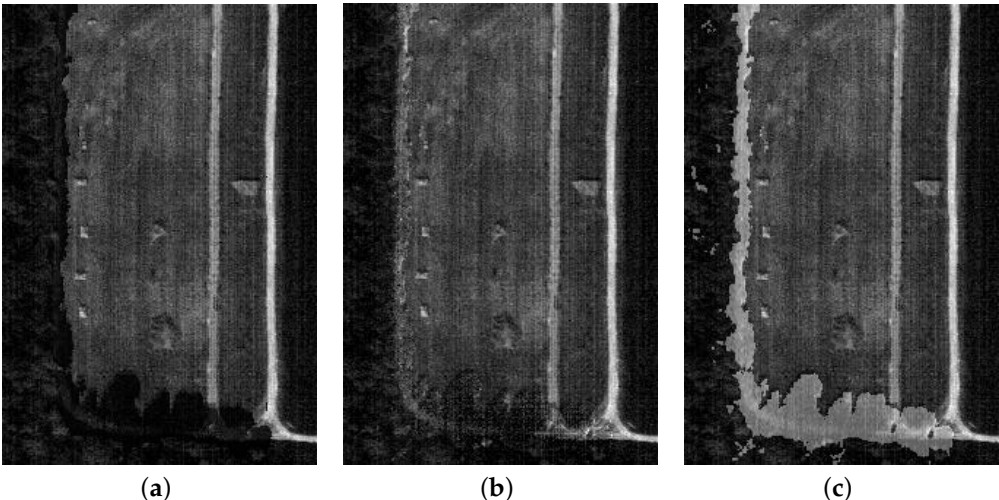

|        (a)        |        (b)        |        (c)        |

**Figure 12.** The $1^{ST}$ band of HYDICE enhanced by 1D and 2D DSR. (**a**) Original HYDICE; (**b**) 1D DSR; (**c**) 2D DSR.

**Table 4.** Classification accuracy of different data by the 3DCNN.

| Data | Original | Enhanced by 1D DSR | Enhanced by 2D DSR |
|---|---|---|---|
| OA | 96.5388 | 97.0277 | 97.3508 |
| AA | 89.8357 | 89.8990 | 91.0473 |
| Kappa | 94.0936 | 94.7531 | 95.4368 |

Compared to the original HYDICE in Figure 12a, 1D DSR can help promote the information expression of the HSI and the ground objects were clearer than before visually. However, the image of the 2D-DSR-enhanced data in the first dimension had lighter brightness, which reduced the impact of the shadow to a greater extent. The effect can be verified as well in the information extraction experiment by the 3DCNN. As shown in Table 4, compared with the classification results of the original data and the 1D-DSR-enhanced data, the application of 2D DSR increased the OA by 0.812% and 0.3231%, the AA by 1.2116% and 1.1483%, and Kappa by 1.3432% and 0.6837%. The application of 1D DSR had a positive effect in improving the information expression, and the classification accuracy can be promoted. Whether in the evaluation of the OA, AA, or Kappa coefficient, the information extraction effect for 2D-DSR-enhanced data was better, which proved the superiority of 2D DSR in spatial information utilization and had great performance in HSI shadow enhancement.

### 3.3.2. Classification Results

To concentrate on the improvement of the classification methods based on the CNN, the 2D and 3DCNN combined with different attention mechanisms often used for the CNN, for instance the squeeze-and-excitation module (SE) [48], global attention block (GAB) [49], dual attention (DA) [50], double ECA (DECA), and CBAM, were compared, and the classification results of the considered methods are listed in Table 5 and Figure 13. Compared with the other methods, the OA and Kappa values achieved the best effect, increasing the OA value by 1.2441%, 1.2369%, 0.9498%, 0.319%, 0.2223%, 0.3358%, 0.4234%, 0.1952%, 0.1677%, 0.1364% and Kappa value by 2.1509%, 2.1274%, 1.1969%, 0.5421%, 0.3838%, 0.2307%, 0.5743%, 0.7304%, 0.4032%, 0.4009%, and 0.2139%, and the AA value of the proposed technique was 90.9980%, only next to that of the 3DCNN. By observing the recall of the compared methods, the proposed method had a better effect for most labels' classification, especially for Target 1, Target 2, and Target 3, the recall values being 0.88%, 4.43%, and 2.86% higher than the 3DCNN.

**Table 5.** Classification accuracy of the considered methods. The 2DCNN and 3DCNN with different attention modules are included.

| Method | 2D CNN | GAB-2DCNN | MAM-2DCNN | 3D CNN | GAB-3DCNN | CBAM-3DCNN | SE-3DCNN | DA-3DCNN | ECA-3DCNN | DECA-3DCNN | ECA-CBAM-3DCNN | MAM-3DCNN |
|---|---|---|---|---|---|---|---|---|---|---|---|---|
| Grass | 0.9900 | 0.9900 | 0.9900 | 0.9900 | 0.9900 | 0.9900 | 0.9900 | 0.9900 | 0.9900 | 0.9900 | 0.9900 | **0.9900** |
| Tree | 0.9725 | 0.9700 | 0.9750 | 0.9820 | 0.9840 | 0.9860 | 0.9820 | 0.9800 | 0.9860 | 0.9860 | 0.9855 | **0.9869** |
| Road | 0.9550 | 0.9600 | 0.9645 | 0.9700 | 0.9740 | 0.9720 | **0.9740** | 0.9680 | 0.9680 | 0.9690 | 0.9695 | 0.9700 |
| Road in shadow | 0.7925 | 0.8260 | 0.8320 | **0.8820** | 0.8420 | 0.8620 | 0.8560 | 0.8560 | 0.8520 | 0.8520 | 0.8430 | 0.8538 |
| Grass in shadow | 0.8975 | 0.8960 | 0.9020 | **0.9440** | 0.9220 | 0.9180 | 0.9240 | 0.9160 | 0.9280 | 0.9120 | 0.9120 | 0.9138 |
| Target 1 | 0.8425 | 0.8760 | 0.8946 | 0.8920 | 0.8760 | 0.8940 | 0.8840 | 0.8720 | 0.8840 | 0.8856 | 0.8860 | **0.9008** |
| Target 2 | 0.6625 | 0.6920 | 0.7220 | 0.7180 | 0.6260 | 0.7060 | 0.6340 | 0.6020 | 0.6420 | 0.6650 | 0.7130 | **0.7623** |
| Target 3 | 0.8775 | 0.8520 | 0.8600 | 0.9060 | 0.9320 | 0.9320 | 0.9200 | 0.9180 | 0.9260 | 0.9200 | 0.9230 | **0.9346** |
| OA (%) | 96.4257 | 96.4329 | 96.7200 | 97.3508 | 97.4475 | 97.5317 | 97.3340 | 97.2464 | 97.4746 | 97.5021 | 97.5334 | **97.6698** |
| AA (%) | 87.2958 | 88.3053 | 88.5750 | **91.0473** | 89.3490 | 90.7666 | 89.6266 | 88.7892 | 89.8707 | 89.8806 | 90.1023 | 90.9980 |
| Kappa (%) | 93.8280 | 93.8515 | 94.4820 | 95.4368 | 95.5951 | 95.7482 | 95.4046 | 95.2485 | 95.5757 | 95.5780 | 95.7650 | **95.9789** |

According to the values of the evaluation indices in Table 5, it can be known that the classification accuracy of 2DCNN-based methods was not as good as that of 3DCNN-based methods. Similar inferences can be drawn from the classification results in Figure 13, especially for the "Road in shadow" pixels. In Figure 13a, some pixels belonging to "Road in shadow" were misclassified into "Tree in shadow" in orange color, because the 2DCNN can only use the spatial information of images, but not the spectral information of HSIs, which plays a vital role in identifying target categories. With the introduction of the attention mechanisms to the 2DCNN, more pixels of "Road in shadow" in Figure 13b,c were correctly classified than in Figure 13a. The 3DCNN can fully utilize the three-dimensional tensor properties of HSIs, which include both spatial and spectral information, so its classification performance Figure 13d is superior to the 2DCNN. Compared with the results in Figure 13 from (d) to (l), for pixels in the lower-right corner, the classification of the 3DCNN was still rough, which indicated the effectiveness of the attention modules. Moreover, due to the single channel attention concentration, the OA values of the GAB-3DCNN, SE-3DCNN, and ECA-3DCNN were lower than 97.50%, and the CBAM combining both channel and spatial attention can better help improve the classification than the former modules. In addition, hundreds of dimensions of spectral data in HSIs lead to complex channel states in the network, so double ECAs, as in Figure 13j, can be more effective at learning channel attention than one ECA, as in Figure 13i, especially when they are combined with the CBAM, as in Figure 13l; the effect is significant.

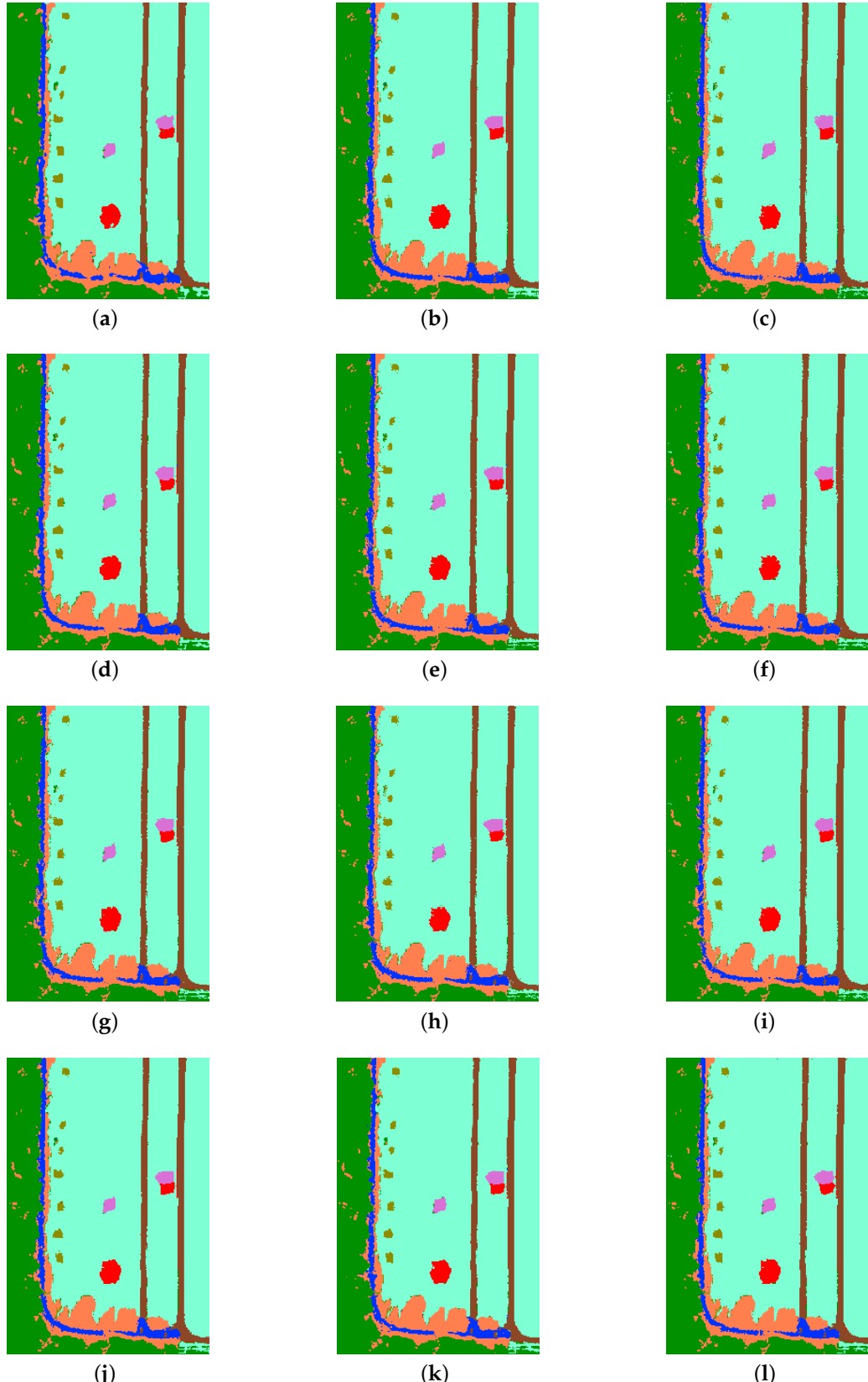

**Figure 13.** Classification results: (**a**) 2DCNN; (**b**) GAB-2DCNN; (**c**) MAM-2DCNN; (**d**) 3DCNN; (**e**) GAB-3DCNN; (**f**) CBAM-3DCNN; (**g**) SE-3DCNN; (**h**) DA-3DCNN; (**i**) ECA-3DCNN; (**j**) DECA-3DCNN; (**k**) ECA-CBAM-3DCNN; (**l**) MAM-3DCNN.

## 4. Discussion

### 4.1. Analysis of 2D DSR Effect on Shadow Enhancement

As shown in Figure 14, the spectral curves of road and grass in the shadow area before and after 2D DSR enhancement are plotted, respectively. It can be observed that the enhancement effect was obvious; specifically, the spectral curve of enhanced grass showed a similar trend for non-shadow grass in Figure 14b.

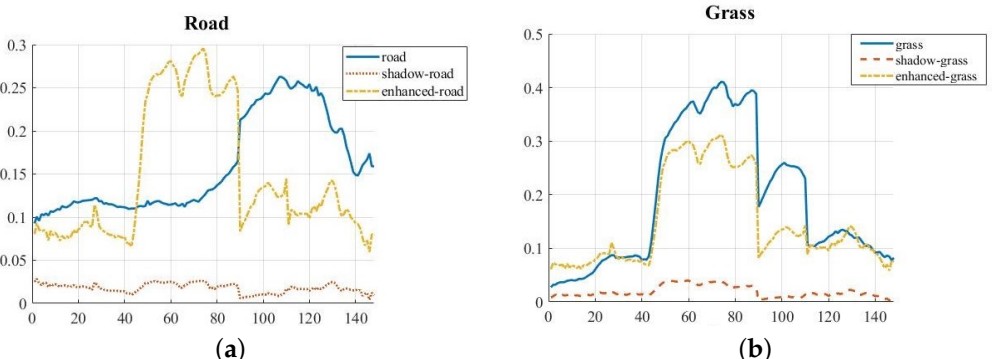

**Figure 14.** Comparison of spectral curves. Solid, dashed, and dotted lines represent the spectral curves of road and grass not in the shadow area, in the shadow areas, and after enhancement, respectively. (**a**) Spectral curves of road. (**b**) Spectral curves of grass.

Compared with the original HYDICE image in Figure 12a, 2D DSR can significantly enhance the shadow area, especially in the 1~40 bands with Figure 14 observed at the same time, while 1D DSR had a relatively weak enhancement effect. Table 4 shows that the application of DSR can effectively promote the classification performance by improving the ability of image feature expression. The proposed 2D DSR improved the OA, AA, and Kappa values by 0.8120%, 1.2116%, and 1.3432%, respectively, which were 0.3231%, 1.1483%, and 0.6837% higher than the classification accuracy of 1D DSR, indicating the positive impact of 2D DSR on information mining.

### 4.2. The Classification Performance Discussion of Considered Measures

As shown in Table 5, benefiting from large samples, all considered methods had a good performance on grass classification. Compared to the 2DCNN and 3DCNN, the attention mechanism had a particular effect of improving the network's performance. Because the 3DCNN can make full use of the 3D data characteristics of HSIs, the classification effect of the method based on the 3DCNN was better than that based on the 2DCNN. Although the GAB-3DCNN, 3DCNN, and ECA-3DCNN performed better on road, road in shadow, and grass in shadow than the other methods, with an overview of all labels, the classification effect of the MAM-3DCNN was prominent. In addition, the MAM module improved the OA and Kappa values of the 3DCNN by 0.3190% and 0.5421%. According to the classification accuracy of the different methods shown in Figure 15, the proposed method performed better than the other considered methods under the evaluation of the OA, AA, and Kappa, among which the AA values were the lowest. Only the AA values of the 3DCNN and MAM-3DCNN were close, but the difference between the AA value of the MAM-3DCNN and the best AA value was only 0.0493%.

Furthermore, according to the classification accuracy shown in Table 5, ECA had a certain effect on extracting the channel attention by increasing the influence of the feature map with a large effect, with the OA and Kappa values improving by 0.1238% and 0.1389%. With the channel and spatial attention combined, the CBAM also had a good function in improving the OA and Kappa values of the 3DCNN by 0.1809% and 0.3114%. The MAM-3DCNN combines the advantages of both the ECA and CBAM attention mechanisms, which can enlarge the impact of key feature maps to a greater extent through dual-channel

attention and spatial attention. The pixels of most categories can be classified by the MAM-3DCNN more accurately than the other considered methods, especially for the small samples Targets 1, 2, and 3, whose OA values improved by 0.88%, 4.43%, and 2.86%, respectively.

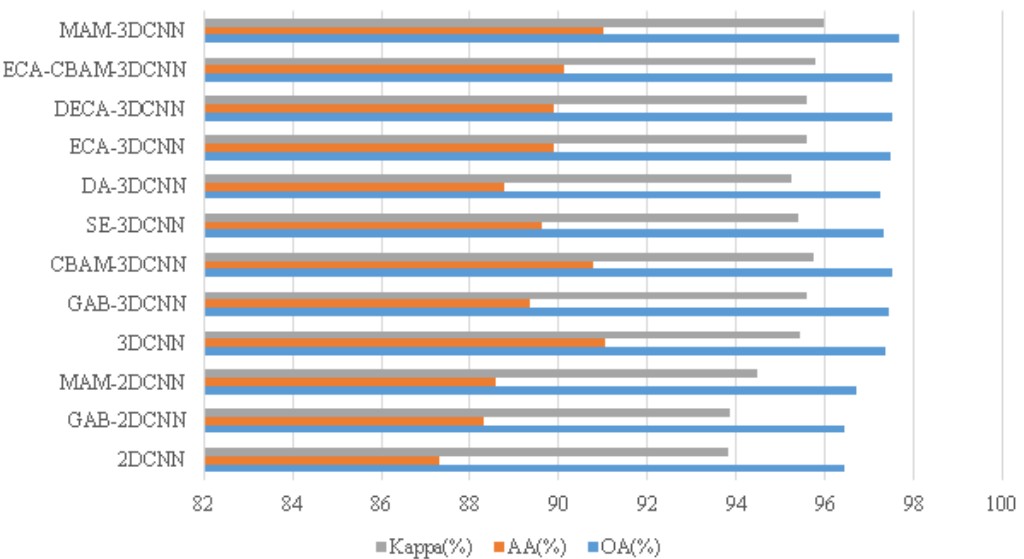

**Figure 15.** The values of the OA, AA, and Kappa of the considered methods.

It can be observed in Figure 13 that most of the considered methods were not good at the classification of road in shadow in the lower left, but the error classification rate of the MAM-3DCNN was lower than the other methods. From Table 5 and Figure 13, the 3DCNN embedded ECA, specially the MAM-3DCNN, can effectively classify labels with few samples, such as grass in shadow, Target 1, and Target 2. Except for the MAM-3DCNN, other methods based on the 3DCNN misclassified some of the leftmost pixels belonging to the tree as grass, as shown in Figure 13e–i. Therefore, the proposed MAM-3DCNN method can help to improve the classification accuracy for enhanced HSIs.

The comparison with the methods based on the GNN [51] and GAN [21] is shown in Table 6. Owing to the function of 2D DSR shadow enhancement, the OA values of the 3D-GAN, MARP-GNN, and MAM-3DCNN improved by 0.8%, 0.94%, and 0.84%. It can be seen that the MAM-3DCNN performed better than the methods in Table 6 because the GNN is designed for irregular data processing and the GAN takes advantage of overcoming the difficulty of insufficient samples. In this paper, the HYDICE data were compatible with the multi-attention combined CNN classification.

**Table 6.** Classification of the original and 2D-DSR-enhanced HYDICE by methods based on the GNN and GAN.

| Evaluation | 3D-GAN | | MARP-GNN | | MAM-3DCNN | |
|---|---|---|---|---|---|---|
| | HYDICE | 2D DSR | HYDICE | 2D DSR | HYDICE | 2D DSR |
| OA (%) | 96.22 | 97.02 | 96.44 | 97.38 | 96.83 | 97.67 |
| AA (%) | 87.33 | 90.13 | 88.50 | 90.35 | 89.61 | 91.00 |
| Kappa (%) | 93.49 | 94.36 | 94.55 | 95.13 | 94.52 | 95.98 |

Owing to the function of 2D DSR shadow enhancement, the results showed that the OA was improved by 0.8%, 0.94%, and 0.84% under the classification of the 3D-GAN, MARP-GNN, and MAM-3DCNN, respectively, which indicated that the more spatial information utilized by the 2D DSR proposed in this paper had a certain effect on the HSI information improvement and the classification accuracy can be promoted. Compared to

the results of the methods based on the GAN and GNN, the MAM-3DCNN had better performance as well. The design of the GNN makes it better at irregular data processing, and the GAN has more advantage by overcoming the difficulty of insufficient samples. In this paper, the adopted data had better compatibility with the multi-attention combined CNN feature extraction.

For the 2DCNN, spectral information from HSIs is not considered, which is crucial for target classification, although attention mechanisms can improve its classification accuracy to some extent. Due to the ability of using both the spectral and spectral dimensional characterists of the HSI data, the 3DCNN is suitable for processing HSIs. The incorporation of the attention mechanism, especially the embedding of double ECAs and CBAM, further improved the classification performance of the 3DCNN.

## 5. Conclusions

Due to complex environmental factors, there are shadow areas in some HSIs, which negatively affect the HSIs' classification. Meanwhile, features extracted from most classification networks have much redundancy. Therefore, a shadow enhancement method based on 2D DSR and a classification model combining a CNN with multiple attention mechanisms for HSIs were proposed in this paper. Firstly, to maintain the spatial correlation of HSIs, an iterative equation of 2D DSR was derived. Next, the weak signal in the shadow area can be increased by 2D DSR, and enhanced HSIs can be obtained. Then, a 3DCNN embedded with two ECA modules and one CBAM was designed to utilize the key features significantly affecting the classification accuracy. Finally, a real-world HSI was used to estimate the performance of the proposed technique. The numerical results showed that 2D DSR outperformed 1D DSR in shadow enhancement of HSIs and the MAM-3DCNN had more competitive classification ability than other considered methods. By applying 2D DSR, the signals and the image quality in the shadow area can be improved. In addition, the HSI classification performance was upgraded by introducing multiple channel and spatial attention modules to the 3DCNN. Therefore, the proposed technique has potential prospects in the shadow information exploration of HSIs.

Although 2D DSR is convenient for processing the image matrix, the 3D tensor characteristic of HSIs was not taken into account. Therefore, these results encourage us to further expand DSR to 3D to protect the 3D data features of HSIs.

**Author Contributions:** Conceptualization, X.L.; data curation, Q.L. and M.F.; formal analysis, X.L.; funding acquisition, X.L. and M.F.; investigation, Q.L., X.L. and M.F.; methodology, X.L. and Q.L.; project administration, X.L. and M.F.; resources, X.L. and Q.L.; software, Q.L. and X.L.; supervision, X.L. and M.F.; validation, X.L., Q.L. and M.F.; visualization, Q.L. and X.L.; writing—original draft, Q.L. and X.L.; writing—review and editing, Q.L., X.L. and M.F. All authors have read and agreed to the published version of the manuscript.

**Funding:** This research was funded by the National Natural Science Foundation of China (Grant No. 61971244) and the Shandong Provincial Natural Science Foundation (Grant No. ZR2020MF011).

**Data Availability Statement:** Not applicable.

**Acknowledgments:** The authors would like to thank the Editors and Reviewers for their valuable comments.

**Conflicts of Interest:** The authors declare no conflict of interest.

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
