# Peer review of "Shadow Enhancement Using 2D Dynamic Stochastic Resonance for Hyperspectral Image Classification"

_remotesensing, doi:10.3390/rs15071820_

Round 1

Reviewer 1 Report

In the paper, an approach to hyperspectral image classification is proposed.   Comments:
1.    The paper contains expressions that may not be related to the subject or is focused on a superficial problem. For example, what do “features extracted from most classification methods based on deep learning have much redundancy” mean?  The paper does not discuss the redundancies in features, nor a way to handle them. Also, deep learning approaches are not designed to extract features. Hence, the problem should be better highlighted. The redundancy is mentioned only in the abstract and conclusion. Hence, the conclusions are not supported.
2.    Authors have already published a method based on dynamic stochastic resonance (X. Liu, H. Wang, Y. Meng and M. Fu, "Classification of Hyperspectral Image by CNN Based on Shadow Area Enhancement Through Dynamic Stochastic Resonance," in IEEE Access, vol. 7, pp. 134862-134870, 2019, doi: 10.1109/ACCESS.2019.2941872.) The differences should be better highlighted than one sentence on page 2. What novel is introduced? That method should be taken into account in the experiments. It seems working on the same data (Fig 9 is a part of Fig 5 in that paper).
3.    The contributions of this study should be enlisted at the end of Section I before the remaining parts of the paper are described.
4.    Please clearly describe the division of data samples into training and testing subsets. Are they disjoint?
5.    Results reported in Section 3.3.2 are just read from the table. Here, readers can easily indicate the best solution. A thorough analysis should be provided. The subfigures are not commented on. What can we see here? Please share the insight.
6.    The classification problems are not discussed in the paper. What happened?  What enabled the improvement of classification accuracy? The paper lacks thorough discussion. All we got here are indications of the best results in tables. A study on several cases should be included.
7.    The results cannot be replicated by a reader.
8.    The paper requires proofreading (e.g., “as commaonly” – page 10).

Author Response

Shadow Enhancement Using 2D Dynamic Stochastic Resonance for Hyperspectral Image Classification

Qiuyue Liu, Min Fu and Xuefeng Liu*

Dear Reviewer,

Please receive our revised manuscript (ID: remotesensing-2259188).

We appreciate you because you give us the valuable suggestions. Thanks a lot for your review and comments which are very helpful to improve our work.

According to all comments and suggestions, we have tried our best to revise our article thoroughly. For your convenience, the changes have been explained point-by-point, at the same time, the details of the revisions and modified parts are marked with red color in the revised manuscript. This revised manuscript of the paper was read and edited by a native English speaker.

We would like to express our great appreciation to you for comments on our article. Looking forward to hearing from you.

Yours Sincerely,

Xuefeng LIU

E-mail: snowclub@qust.edu.cn

Reviewer 2 Report

Improve abstract. Explain the contribution, promising prospects are not good enough. I do not agree with accuracy as a means to reporting results due to its obvious deficiencies. More explanation on captions of the images/tables. Figure 15 is misleading as connecting points with lines implies connection between points. Is that what the authors wanted to convey? I do not know how to read Figure 15 but kappa seems better than AA. Text says otherwise... Conclusion is underwhelming. 3 selfs.

Reviewer 3 Report

The authors have proposed a a method based on 2-dimensional dynamic stochastic resonance (2D DSR) and convo- 4 lutional neural network (CNN) combined with multiple attention mechanisms for HSI classification. The manuscript is complete, and the authors try to prove the progressiveness of the algorithm through experiments. However, there are some problems that need to be revised. The comments are as follows

1. The motivations or remaining challenges are not so clear or what kinds of issues or difficulties are this task that is facing. Please give more details and discussion about the key problems solved in this paper, which is largely different from existing works.

2. A deep literature review should be given, particularly advanced and SOTA deep learning or AI models in hyperspectral image classification. Therefore, the reviewer suggests discussing some currently SOTA works by analyzing the following papers in the revised manuscript, e.g., Semi-Supervised Locality Preserving Dense Graph Neural Network With ARMA Filters and Context-Aware Learning, Unsupervised Self-correlated Learning Smoothy Enhanced Locality Preserving Graph Convolution Embedding Clustering, Self-supervised Locality Preserving Low-pass Graph Convolutional Embedding, Deep hybrid: multi-graph neural network collaboration for hyperspectral image classification, Multi-feature Fusion: Graph Neural Network and CNN Combining.

3. How about the computational complexity?

4. The compared methods are not sufficient. Some SOTA compared methods should be involved.

5. It is well-known that the hyperspectral image usually tend to suffer from various degradation, noise effects, or variabilities in the process of imaging. Please give the discussion and analysis by referring to the paper titled by e.g., MultiReceptive Field: An Adaptive Path Aggregation Graph Neural Framework. The reviewer is wondering what will happen if the proposed method meets the various variabilities.

Round 2

Reviewer 1 Report

The revision significantly improved the paper.

Reviewer 2 Report

Author answered my concerns.

Reviewer 3 Report

No more comments.